

# Development and validation of an in-hospital major adverse cardiovascular events risk model for young patients with acute coronary syndrome: a retrospective cohort study

Jia Zheng[1], Junyang Li[2], Tingting Li[3], Fang Hu[1], Degang Cheng[1] and Chengzhi Lu[1]

[1] Department of Cardiology, Tianjin First Central Hospital, Tianjin, China
[2] Department of Neurosurgery, Chinese People's Liberation General Hospital, Beijing, China
[3] Department of Respiratory and Critical Care Medicine, Shandong Provincial Hospital Affiliated to Shandong First Medical University, Jinan, China

Corresponding author
Chengzhi Lu,
5020200072@nankai.edu.cn

## ABSTRACT

**Background:** The incidence of acute coronary syndrome (ACS) among young individuals is increasing, making it a leading cause of mortality in this population. This study aimed to develop and validate a risk prediction model for in-hospital major adverse cardiovascular events (MACE) in young ACS patients.

**Methods:** A retrospective analysis was performed to predict in-hospital MACE. Patients were divided into a training set ($n = 342$) and a testing set ($n = 171$). Screening variables were optimized using least absolute shrinkage and selection operator (LASSO) regression and univariable logistic regression analysis. A predictive nomogram model was developed through multivariate logistic regression. The model's discrimination and calibration were assessed using the receiver operating characteristic (ROC) curve, calibration plots, and Hosmer-Lemeshow goodness-of-fit tests. Clinical utility was evaluated using decision curve analysis (DCA).

**Results:** White blood cell count, Killip classification, lymphocyte count, heart rate, triglycerides, and Gensini score were identified as significant predictors. The constructed nomogram demonstrated strong predictive performance. The area under the ROC curve (AUC) was 0.9242 (95% confidence interval [CI]: [0.8841–0.9643]) for the training set and 0.8346 (95% CI [0.742–0.9272]) for the testing set, with respective cut-off values of 0.107 and 0.119. Calibration was confirmed with Hosmer-Lemeshow statistics of 12.454 ($p = 0.2558$) in the training set and 7.16 ($p = 0.7102$) in the testing set. DCA showed threshold probabilities ranging from 0% to 100% in the training set and 0% to 90% in the testing set.

**Conclusions:** The proposed nomogram model demonstrated robust discrimination and calibration, offering a valuable tool for predicting the risk of in-hospital MACE in young ACS patients.

# INTRODUCTION

Young patients account for 4–10% of all people who experience acute coronary syndrome (ACS) (*Doolub, Kandoole-Kabwere & Felekos, 2022*), and this figure has been steadily rising (*Meirhaeghe et al., 2020*). The effect of ACS on the daily lives of young patients is distressing, and they pose a substantial economic and health care burden on society, often transitioning into chronic patients (*Page et al., 2013*). A large cohort study found that the 30-day mortality rate for acute myocardial infarction (AMI) undergoing primary percutaneous coronary intervention is as high as 20.7% (*Suryawan et al., 2025*). In young adults with ACS, the overall in-hospital mortality rate is 3.3% (*Tsai et al., 2017*). Patients under 45 years of age with ST-segment elevation myocardial infarction (STEMI) have an in-hospital mortality rate of approximately 1% (*Rathod et al., 2015*). Despite advancements in coronary artery disease (CAD) treatment and secondary prevention, there has been little improvement in the overall incidence of adverse cardiovascular events in younger patients (*Han et al., 2019*). For such patients, the prediction of major adverse cardiovascular events (MACE) has a significant impact on medical decision-making and the selection of treatment. As a result, it is critical to identify early risk classifications and timely interventions to improve the prognosis of young ACS patients.

Although several risk factors for MACE exist in young ACS patients (*DeFilippis et al., 2015*; *Boos, Toon & Almahdi, 2021*), the specific predictive value of each factor remains unclear, limiting its clinical application. Therefore, developing a predictive model is urgently needed. Such a model may be used to help assess the in-hospital MACE risk for young ACS patients and guide early treatment responses for better outcomes. Additionally, there is little information on predictive models for the occurrence of MACE in ACS patients under 45 years of age.

In summary, the objectives of this study included developing a clinical prediction model, presenting the model through column line plots, and then validating it externally to create personalized treatment plans for young patients with ACS according to their in-hospital MACE risk.

# MATERIALS AND METHODS

## Study population and design

This retrospective study included 513 young ACS patients between January 2018 and July 2024. The ACS diagnostic criteria followed the guidelines of the American College of Cardiology for patients with STEMI, non-STEMI, and unstable angina (UA).

The study design strictly adhered to the Transparent Reporting of a Multivariable Prediction Model for Individual Prognosis or Diagnosis (TRIPOD) guidelines (*Collins et al., 2015*), specifically meeting the event-per-variable (EPV) ratio requirement of ≥10 (*Harrell, 2010*; *Riley et al., 2020*). In the training set ($n = 342$), we ensured an EPV ratio of ≥10 for each of the 26 candidate predictors, thereby exceeding the minimum stability

threshold necessary for reliable parameter estimation. Furthermore, the allocation of the testing set (33.3% of the total sample, $n = 171$) followed methodological recommendations by preserving a 4:1 training-to-validation ratio to optimize model generalizability and ensuring sufficient statistical power for external validation through variance reduction strategies (*Molinaro, Simon & Pfeiffer, 2005*; *Kim, 2009*; *Bischl et al., 2012*).

Subsequently, we performed least absolute shrinkage and selection operator (LASSO) regression using the R package glmnet (*Friedman, Hastie & Tibshirani, 2010*), a technique that incorporates L1 regularization to facilitate automated variable selection. Our analysis included 26 candidate predictors, whose penalized coefficient estimates were progressively shrunk to zero as the regularization parameter $\lambda$ increased, based on their relative contributions to model performance. The complexity-tuning parameter $\lambda$ was determined through 10-fold cross-validation resampling. We adopted the $\lambda.1se$ criterion as the final regularization parameter, which corresponds to the largest $\lambda$ value within one standard error of the minimum cross-validation error ($\lambda.min$). This approach achieves an optimal trade-off between predictive accuracy and model parsimony by selecting the simplest model structure that maintains statistically equivalent predictive performance.

In our training set, the application of the $\lambda.1se$ penalty retained only a subset of predictors with non-zero coefficients, which were subsequently incorporated into the final parsimonious prediction model. We then conducted subsequent univariate and multivariate logistic regression analyses. The significant predictors identified through multivariate analysis were incorporated into a personalized prediction model for MACE. A nomogram was constructed to visually represent the predictive model. Based on the multivariate logistic regression coefficients, each predictor was assigned specific weights to calculate individual risk scores. This predictive model was evaluated based on three main aspects: discrimination, calibration, and net clinical benefit (*Friedman, Hastie & Tibshirani, 2010*; *Tay, Narasimhan & Hastie, 2023*).

The study adhered to the Helsinki Declaration, and approval was obtained from the Tianjin First Central Hospital (approval No. 20240515-1). The flowchart of our study was shown in Fig. 1.

## Inclusion criteria and exclusion criteria

Inclusion criteria were as follows:

1. Patients met the diagnostic criteria of ACS (*O'Gara et al., 2013*; *Amsterdam et al., 2014*) and received coronary angiography (CAG);
2. 18 years < patient age < 45 years (*Sawada et al., 2020*; *Chinese Society of Cardiology, Chinese Medical Association & Editorial Board of Chinese Journal of Cardiology, 2025*);
3. Clinical data and ultrasonography were completed.

The exclusion criteria were as follows:

1. Patients with a prior history of ACS.
2. Patients with heart failure, infection, hyperthyroidism, liver and kidney dysfunction, hematological disorders, and cerebrovascular diseases on admission.

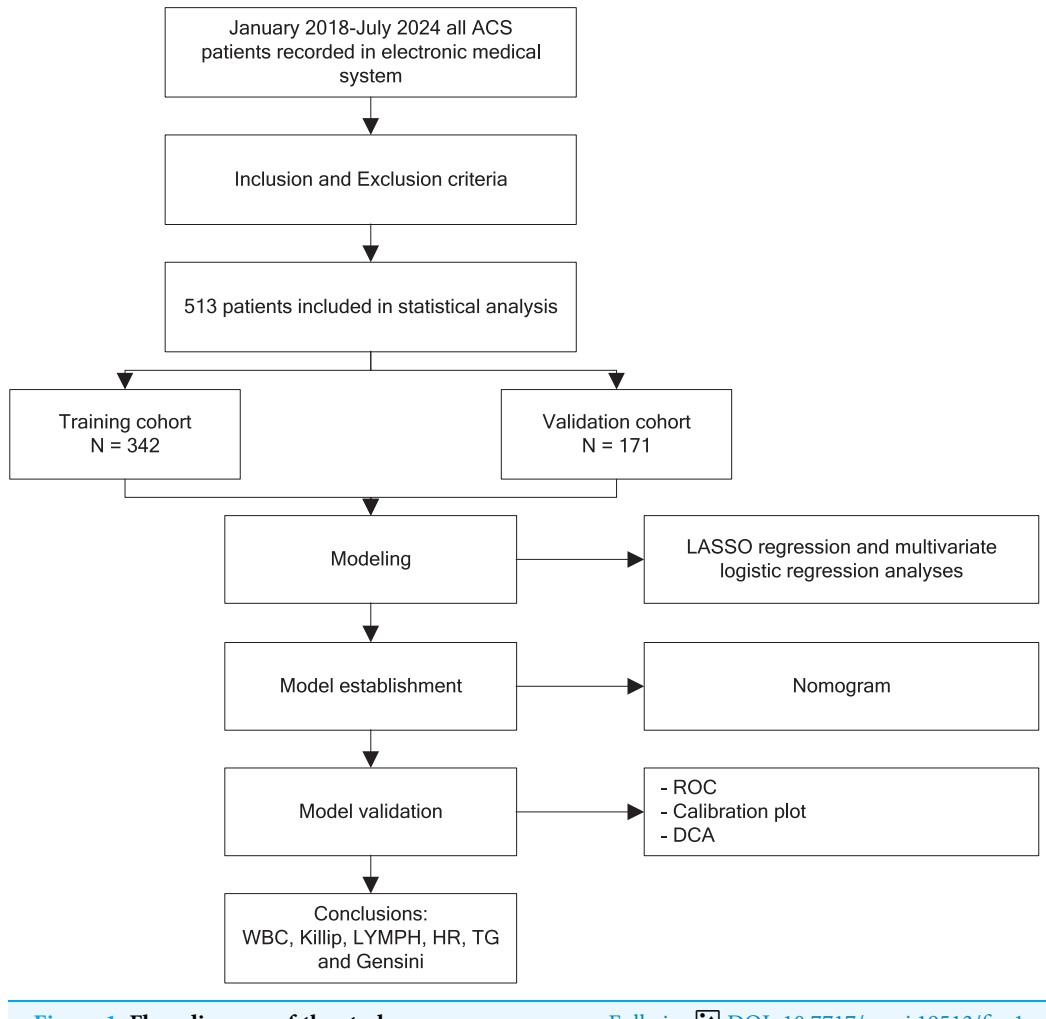

**Figure 1** Flow diagram of the study.

3. Other heart disease patients (including severe congenital heart disease, valve disease, cardiomyopathy, and myocarditis).

4. The patients without CAG.

## Clinical data collection

Clinical data were collected through electronic medical record systems. The datasets encompassed baseline demographic characteristics, clinical presentations, comorbid conditions, laboratory parameters, echocardiographic indices, and angiographic metrics. Blood samples were taken upon admission. Detailed indicators are as follows:

1. Demographic and basic clinical characteristics: age, sex, smoking habits, history of hypertension and diabetes, and family history.

2. Inflammation and hematologic markers: neutrophil count (Neu), lymphocyte count (LYMPH), white blood cell count (WBC), and the neutrophil/lymphocyte ratio (NLR) were calculated.

**Table 1 Clinical characteristics of the study population.**

| Characteristics | Overall N = 513 | Training N = 342 | Testing N = 171 | P | NO-MACE N = 446 | MACE N = 67 | P |
|---|---|---|---|---|---|---|---|
| Clinical presentation | | | | 0.673 | | | <0.001 |
| STEMI | 306 (59.6%) | 202 (59.1%) | 104 (60.8%) | | 248 (55.6%) | 58 (86.6%) | |
| NSTEMI | 98 (19.1%) | 69 (20.2%) | 29 (17.0%) | | 89 (20.0%) | 9 (13.4%) | |
| Unstable angina | 109 (21.2%) | 71 (20.8%) | 38 (22.2%) | | 109 (24.4%) | 0 (0.00%) | |
| Sex (%) | | | | 0.346 | | | 0.121 |
| Male | 488 (95.1%) | 328 (95.9%) | 160 (93.6%) | | 427 (95.7%) | 61 (91.0%) | |
| Female | 25 (4.87%) | 14 (4.09%) | 11 (6.43%) | | 19 (4.26%) | 6 (8.96%) | |
| Smoke (%) | 343 (66.9%) | 224 (65.5%) | 119 (69.6%) | 0.407 | 295 (66.1%) | 48 (71.6%) | 0.452 |
| Diabetes mellitus (%) | 84 (16.4%) | 48 (14.0%) | 36 (21.1%) | 0.058 | 67 (15.0%) | 17 (25.4%) | 0.05 |
| Hypertension (%) | 180 (35.1%) | 111 (32.5%) | 69 (40.4%) | 0.095 | 159 (35.7%) | 21 (31.3%) | 0.581 |
| History (%) | 91 (17.7%) | 62 (18.1%) | 29 (17.0%) | 0.838 | 80 (17.9%) | 11 (16.4%) | 0.895 |
| Killip classification (%) | | | | 0.26 | | | <0.001 |
| I | 454 (88.5%) | 307 (89.8%) | 147 (86.0%) | | 421 (94.4%) | 33 (49.3%) | |
| ≥II | 59 (11.5%) | 35 (10.2%) | 24 (14.0%) | | 25 (5.61%) | 34 (50.7%) | |
| Age, years ([IQR]) | 38.0 [35.0; 42.0] | 38.0 [34.0; 42.0] | 39.0 [35.0; 42.0] | 0.255 | 38.0 [34.0; 42.0] | 38.0 [35.0; 42.0] | 0.832 |
| SBP | 136 (22.3) | 134 (23.1) | 138 (20.5) | 0.059 | 137 (21.4) | 129 (26.8) | 0.028 |
| DBP | 88.6 (16.6) | 87.8 (16.8) | 90.0 (16.0) | 0.153 | 89.2 (15.9) | 84.3 (20.2) | 0.06 |
| HR | 78.0 [68.0; 87.0] | 78.0 [68.0; 87.0] | 80.0 [69.0; 87.0] | 0.544 | 77.5 [68.0; 86.0] | 85.0 [70.5; 99.5] | 0.001 |
| Neutrophil (×10$^9$/L) ([IQR]) | 6.50 [4.89; 9.15] | 6.40 [4.81; 9.19] | 6.80 [5.25; 9.05] | 0.21 | 6.37 [4.71; 8.52] | 9.19 [6.88; 12.4] | <0.001 |
| WBC (×10$^9$/L) ([IQR]) | 9.92 [7.84; 12.2] | 9.72 [7.74; 11.9] | 10.1 [8.20; 12.5] | 0.251 | 9.50 [7.73; 11.6] | 13.4 [9.97; 16.5] | <0.001 |
| LYMPH (×10$^9$/L) ([IQR]) | 2.05 [1.47; 2.82] | 2.05 [1.57; 2.84] | 2.03 [1.42; 2.78] | 0.514 | 2.05 [1.49; 2.76] | 2.01 [1.44; 3.13] | 0.735 |
| NLR ([IQR]) | 2.96 [1.98; 5.42] | 2.93 [1.95; 5.12] | 3.07 [2.06; 6.25] | 0.278 | 2.79 [1.97; 4.78] | 4.89 [2.40; 7.35] | <0.001 |
| Gensini ([IQR]) | 43.0 [25.0; 69.0] | 42.0 [24.0; 67.0] | 48.0 [29.0; 70.5] | 0.264 | 41.0 [22.5; 63.0] | 80.0 [50.0; 105] | <0.001 |
| CK-MB(U/L)[IQR] | 3.06 [1.40; 17.9] | 2.85 [1.40; 17.6] | 3.88 [1.50; 21.0] | 0.244 | 2.87 [1.40; 17.2] | 4.80 [1.60; 32.0] | 0.212 |
| Uric acid (umol/L) ([IQR]) | 381 [312; 456] | 376 [312; 455] | 394 [306; 458] | 0.621 | 379 [312; 454] | 402 [296; 469] | 0.569 |
| BUN (mmol/L) ([IQR]) | 5.17 [4.27; 5.99] | 5.06 [4.27; 5.98] | 5.19 [4.30; 6.00] | 0.643 | 5.17 [4.27; 5.99] | 5.02 [4.24; 6.11] | 0.9 |
| Scr (umol/L) ([IQR]) | 73.4 [64.0; 83.0] | 73.0 [64.0; 83.0] | 74.0 [63.5; 82.8] | 0.643 | 73.3 [64.0; 82.5] | 73.5 [65.5; 86.4] | 0.726 |
| Blood glucose ([IQR]) | 6.51 [5.39; 8.90] | 6.47 [5.33; 8.88] | 6.51 [5.66; 8.96] | 0.347 | 6.47 [5.37; 8.90] | 6.83 [5.73; 9.07] | 0.258 |
| TG (mmol/L) ([IQR]) | 2.06 [1.41; 3.40] | 2.04 [1.41; 3.37] | 2.14 [1.39; 3.54] | 0.659 | 2.03 [1.39; 3.37] | 2.43 [1.52; 4.45] | 0.047 |
| LDL-C (mmol/L) ([IQR]) | 3.31 [2.67; 3.96] | 3.27 [2.65; 3.95] | 3.38 [2.70; 3.96] | 0.675 | 3.32 [2.69; 3.93] | 3.31 [2.63; 4.08] | 0.896 |
| HDL-C (mmol/L) ([IQR]) | 0.94 [0.80; 1.09] | 0.94 [0.82; 1.08] | 0.95 [0.79; 1.11] | 0.858 | 0.94 [0.81; 1.10] | 0.94 [0.78; 1.04] | 0.228 |
| TC (mmol/L) ([IQR]) | 4.93 [4.23; 5.72] | 4.92 [4.17; 5.76] | 4.96 [4.28; 5.65] | 0.68 | 4.90 [4.25; 5.71] | 5.13 [4.21; 5.92] | 0.272 |
| LVEF, % ([IQR]) | 58.0 [54.0; 60.0] | 58.0 [55.0; 60.0] | 58.0 [52.5; 60.0] | 0.564 | 58.0 [54.0; 60.0] | 58.0 [53.5; 60.0] | 0.665 |
| MACE (%) | | | | 0.247 | | | |
| 0 | 446 (86.9%) | 302 (88.3%) | 144 (84.2%) | | | | |
| 1 | 67 (13.1%) | 40 (11.7%) | 27 (15.8%) | | | | |

**Note:**

SBP, systolic blood pressure; DBP, diastolic blood pressure; HR, heart rate; LVEF, left ventricular ejection fraction; MACE, major adverse cardiac cardiovascular events; WBC, white blood cell; NLR, neutrophil to lymphocyte ratio; TG, triglycerides; LDL-C, low-density lipoprotein cholesterol; HDL-C, high-density lipoprotein cholesterol; TC, total cholesterol; Scr, creatinine; BUN, urea nitrogen; CK-MB, creatine kinase isoenzyme; LYMPH, lymphocyte.

3. Cardiac function and hemodynamic indices: heart rate (HR), blood pressure (BP), and cardiac color Doppler ultrasound results, including left ventricular ejection fraction (LVEF) and Killip classification.

4. Metabolic and biochemical indicators: triglycerides (TG), total cholesterol, high density lipoprotein (HDL) cholesterol, low density lipoprotein (LDL) cholesterol, blood glucose, blood urea nitrogen (BUN), and creatinine levels.

5. Other key indicators: cardiac enzymes such as creatine kinase isoenzyme (CK-MB).

6. Gensini score: This score was calculated to assess the severity of coronary artery lesions (*Gensini, 1983*).

The basic characteristics and laboratory parameters were not significantly different between the two cohorts, as displayed in Table 1.

## Observed outcomes

We also recorded the incidence of MACE during hospitalization, including cardiac death, recurrent myocardial infarction, severe malignant arrhythmia and heart failure, stroke (hemorrhagic and ischemic), and cardiogenic shock.

The specific evaluation criteria based on guidelines and consensus were as follows:

1. Cardiac death (*Task Force Members et al., 2013*; *Byrne et al., 2023*): autopsy-confirmed myocardial necrosis or coronary occlusion, or clinical exclusion of non-cardiac causes.

2. Re-infarction (*Thygesen et al., 2018*): chest pain lasting for 20 min or more, accompanied by an elevation of high-sensitivity cardiac troponin T of 20% or more from baseline, or new ischemic Q-waves or imaging evidence.

3. Severe arrhythmia (*Al-Khatib et al., 2018*): sustained ventricular tachycardia, ventricular fibrillation, cardiac arrest, or hemodynamically unstable third-degree atrioventricular block.

4. Heart failure (*McDonagh et al., 2021*): the presence of clinical symptoms such as dyspnea or pulmonary edema, and signs such as lung rales or edema, or LVEF less than or equal to 40%.

5. Stroke (*Powers et al., 2019*; *Gil-Garcia et al., 2022*): neurological deficits lasting more than 24 h with CT or MRI-confirmed infarction or hemorrhage.

6. Cardiogenic shock (*Kapur et al., 2022*): systolic BP less than 90 mmHg for more than 30 min requiring vasopressors, cardiac index less than 2.2 L/min/m², or mechanical circulatory support for cardiogenic shock.

Protocol-trained researchers performed standardized MACE adjudication with dual-entry verification of electronic medical record-derived data, ensuring endpoint consistency through applied definitions and criteria, while maintaining accuracy and completeness.

## Sample size and data preprocessing

Determining the effective sample size in a prediction study is dependent on the number of outcome events that occur (*Riley et al., 2020*). To ensure the reliability of the experiment

and achieve meaningful outcomes, addressing the issue of sample imbalance is essential. The oversampling of positive cases was conducted in the training set (*Menardi & Torelli, 2012*).

## Statistical analysis

Statistical methodologies were systematically adapted to data characteristics: continuous variables conforming to normal distribution were quantified as mean ± standard deviation and subjected to intergroup analysis *via* Student's t-test. Nonparametric continuous variables were characterized by median (interquartile range, IQR) with comparative assessments conducted using the Mann-Whitney U procedure. Categorical parameters were enumerated as absolute frequencies (%) and evaluated through Pearson's $\chi^2$ contingency tests. During model development, 26 candidate features in the training cohort underwent feature selection through LASSO regression. Variables retained from this process were subsequently analyzed using univariable logistic regression, with those demonstrating statistical significance ($p < 0.05$) progressing to multivariable logistic regression modeling (*Harrell, 2025*). The final prediction model generated regression coefficients, odds ratios, and 95% confidence intervals, which were visualized through a nomogram. Model validation encompassed three dimensions: discriminative capacity: evaluated using receiver operating characteristic curve analysis; calibration precision: assessed through calibration plots and Hosmer-Lemeshow goodness-of-fit testing; clinical applicability: quantified *via* decision curve analysis. All analyses were conducted using R software v4.4.3 (*R Core Team, 2025*). Statistically significant differences were defined as those for which $p < 0.05$ or $0.01$.

## RESULTS

### Baseline patient characteristics

The final analysis included 513 participants, with a median age of 38 years (range: 35–42 years). Of the cohort, 95.1% were male. Non-ST segment elevation ACS (NSTE-ACS) was observed in 19.1% of patients, while 59.6% presented with ST-elevation myocardial infarction (STEMI). The median HR was 78 bpm (range: 68–87 bpm). Patients were assigned to a training set ($n = 342$) and a testing set ($n = 171$), with no significant differences in demographic or laboratory characteristics between the two groups (all $p > 0.05$). The incidence of in-hospital MACE, defined as either absence (MACE = 0) or presence (MACE = 1) of events following CAG, was 15.8% in the testing set and 11.7% in the training set. Key variables, including WBC, NLR, HR, Killip classification, and Gensini score, were significantly higher in the MACE group compared to the no-MACE group ($p < 0.01$) (Table 1).

### LASSO and univariate logistic regression analysis for screening variables

Predictive factors were initially identified using LASSO regression analysis. The regularization path plots illustrated the dynamic evolution of the predictors' coefficients across the λ spectrum. Increasing λ values correspond to progressively stronger

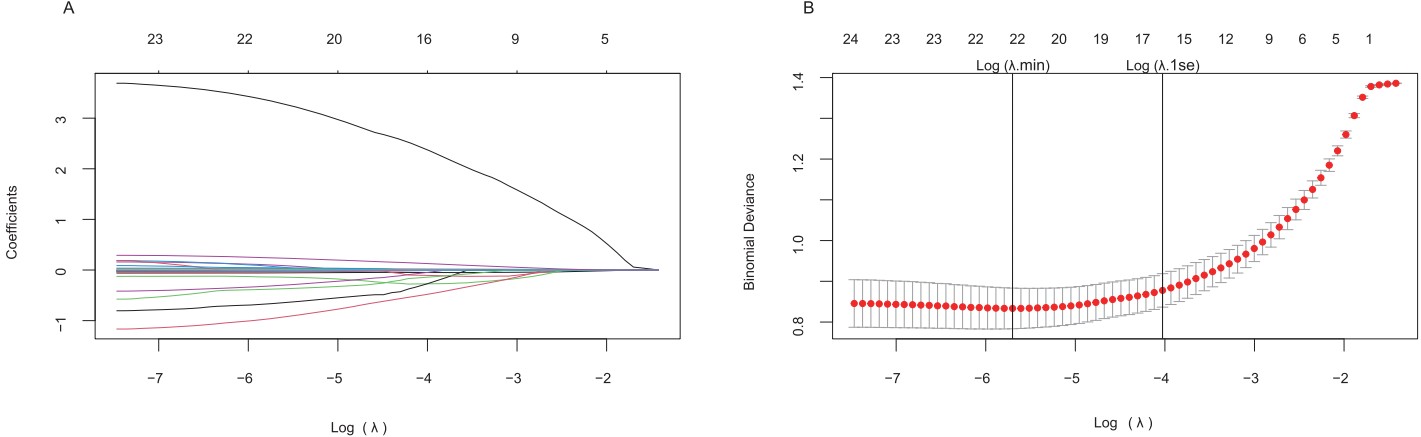

**Figure 2  LASSO regression model for screening predictive factors.** (A) Regularization path plot: The x-axis represents the regularization strength (log(λ)), while the y-axis displays the magnitude of standardized coefficient estimates for predictor variables. As λ increases, the coefficients undergo shrinkage toward zero, with all predictors ultimately converging to zero coefficients under full regularization (λ → ∞). (B) Cross-validation error trajectory plot: The x-axis represents log(λ), and the y-axis quantifies the binomial deviance-a performance metric for the binary classification of MACE occurrence, where lower values indicate better model accuracy. Vertical error bars depict the variability range across cross-validation folds.

regularization penalties. The inflection points of these paths reflect the relative importance of the variables, where steeper declines indicate earlier exclusion from the model during the regularization process (Fig. 2A). The cross-validation error trajectory plot facilitates optimal λ determination by evaluating the bias-variance trade-off across varying regularization intensities. Two critical regularization thresholds are marked; the first vertical black line indicates log(λ.min), while the second black line denotes log(λ.1se). By applying the log(λ.1se) penalty, we selected a restricted subset of predictors demonstrating non-zero coefficients, ultimately establishing a parsimonious predictive model that balances complexity reduction with preserved discriminative capacity (Fig. 2B).

At last, the selected predictors included blood glucose, BUN, CK-MB, Gensini score, HR, Killip classification (Killip I and Killip II–IV), LDL, LYMPH, smoking status, triglyceride (TG), uric acid (UA), and WBC (Table 2).

## Predictive nomogram model development

Based on the predictive factors selected through LASSO regression, we conducted both univariate and multivariate logistic regression analyses to explore the risk factors for in-hospital MACE. The univariate regression analysis revealed that the Gensini score, HR, Killip classification II–IV, LYMPH, TG, and WBC were independent risk factors for in-hospital MACE (odds ratio (OR): 1.02, 95% confidence interval (CI) [1.01–1.03], $p < 0.001$; OR: 1.04, 95% CI [1.02–1.06], $p < 0.001$; OR: 16.17, 95% CI [7.27–35.96], $p < 0.001$; OR: 1.37, 95% CI [1.09–1.72], $p < 0.01$; OR: 1.13, 95% CI [1.03–1.24], $p < 0.01$; and OR: 1.23, 95% CI [1.13–1.33], $p < 0.001$, respectively) (Table 2).

These predictors were incorporated into a multivariate logistic regression model to predict the occurrence of in-hospital MACE. Moreover, the multivariate logistic regression

**Table 2 Univariable and multivariable logistic regression analysis was performed to evaluate the factors that contribute to the occurrence of in-hospital MACE in young ACS patients following CAG.**

| Characteristics | NO-MACE | MACE | OR (univariable) | OR (multivariable) |
|---|---|---|---|---|
| Blood glucose | 7.9 ± 4.0 | 8.3 ± 4.1 | 1.02 (0.95–1.11, $p$ = 0.529) | |
| BUN | 5.2 ± 1.3 | 5.3 ± 1.4 | 1.06 (0.82–1.36, $p$ = 0.668) | |
| CK-MB | 21.6 ± 47.8 | 25.4 ± 47.5 | 1.00 (1.00–1.01, $p$ = 0.635) | |
| Gensini | 45.1 ± 33.6 | 78.4 ± 47.2 | 1.02 (1.01–1.03, $p$ < 0.001) | 1.02 (1.01–1.03, $p$ < 0.001) |
| HR | 77.6 ± 13.5 | 89.4 ± 27.0 | 1.04 (1.02–1.06, $p$ < 0.001) | 1.02 (1.00–1.04, $p$ = 0.106) |
| Killip I | 286 (94.7%) | 21 (52.5%) | | |
| Killip II–IV | 16 (5.3%) | 19 (47.5%) | 16.17 (7.27–35.96, $p$ < 0.001) | 18.69 (6.63–52.71, p < 0.001) |
| LDL-C | 3.4 ± 1.1 | 3.2 ± 1.0 | 0.88 (0.64–1.20, $p$ = 0.416) | |
| LYMPH | 2.3 ± 1.0 | 2.9 ± 2.1 | 1.37 (1.09–1.72, $p$ = 0.007) | 1.18 (0.87–1.59, $p$ = 0.291) |
| Smoke | 102 (33.8%) | 16 (40%) | | |
| | 200 (66.2%) | 24 (60%) | 0.77 (0.39–1.50, $p$ = 0.437) | |
| TG | 2.8 ± 2.5 | 4.1 ± 3.8 | 1.13 (1.03–1.24, $p$ = 0.008) | 1.10 (0.97–1.24, $p$ = 0.139) |
| Uric acid | 387.2 ± 99.2 | 396.8 ± 117.5 | 1.00 (1.00–1.00, $p$ = 0.571) | |
| WBC | 9.9 ± 3.3 | 13.3 ± 4.7 | 1.23 (1.13–1.33, $p$ < 0.001) | 1.19 (1.07–1.33, $p$ = 0.002) |

**Note:**
HR, heart rate; WBC, white blood cell; TG, triglycerides; LDL-C, low-density lipoprotein cholesterol; cholesterol; BUN, urea nitrogen; CK-MB, creatine kinase isoenzyme; LYMPH, lymphocyte.

analysis indicated that the Gensini score, Killip classification II–IV, and WBC were independent risk factors for in-hospital MACE (OR: 1.02, 95% CI [1.01–1.03], $p$ < 0.001; OR: 18.69, 95% CI [6.63–52.71], $p$ < 0.001; and OR: 1.19, 95% CI [1.07–1.33], $p$ < 0.01, respectively) (Table 2).

A nomogram based on the multivariate regression model was developed to predict the outcome of MACE events. The nomogram was constructed as a column line graph (Fig. 3) to visually represent the numerical scores associated with each variable. By assigning the value of each predictive variable to the 'point axis' and summing them to obtain the 'total point score,' we can ultimately map this onto the 'probability axis' to predict the risk of in-hospital MACE (Fig. 3). This allows clinicians to estimate the likelihood of in-hospital MACE for individual patients. Higher total scores indicate a greater risk of MACE, providing valuable guidance for clinical decision-making.

## Validation of the predictive nomogram model

The predictive model demonstrated excellent discriminatory and calibration capabilities. The area under the ROC curve (AUC) for the training set was 0.9242 (95% confidence interval [CI] [0.8841–0.9643]) with a cut-off value of 0.107 (Fig. 4A). For the testing set, the AUC was 0.8346 (95% CI [0.742–0.9272]) with a cut-off value of 0.119 (Fig. 4B). Calibration was assessed using the Hosmer-Lemeshow test, with chi-square statistics of 12.454 ($p$ = 0.2558) for the training set and 7.16 ($p$ = 0.7102) for the testing set. Calibration curves showed strong agreement between predicted and actual probabilities of in-hospital MACE for both sets (Figs. 4C and 4D).

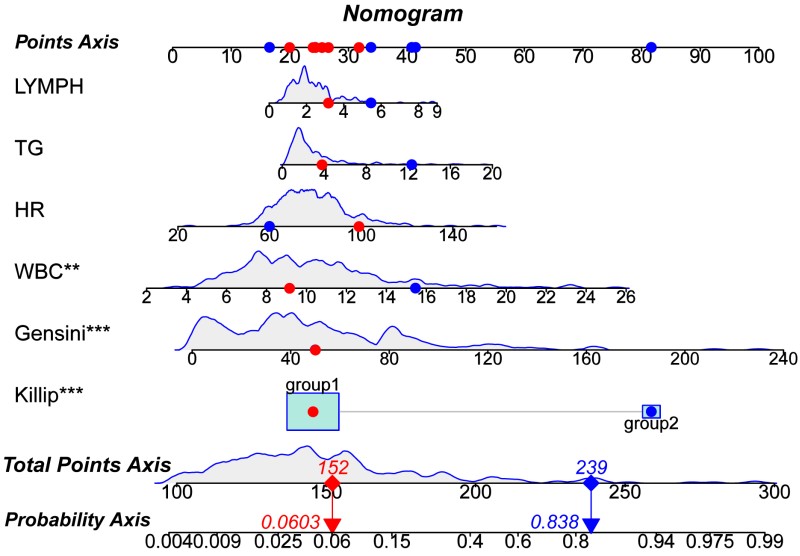

**Figure 3 Nomogram prediction model for in-hospital MACE.** The nomogram illustrates its clinical utility through two example cases: Patient 1 (red dot) has a predicted in-hospital MACE risk of 6.03% (notably, this patient did not experience a MACE event in actuality). In contrast, Patient 2 (blue dot) has a predicted risk of 83.8% (this patient did experience a MACE event in actuality).

## Clinical utility of the predictive nomogram model

Decision curve analysis (DCA) demonstrated the clinical utility of the nomogram. The areas under the black, red, and blue solid lines represented the clinical net benefit, with the red line and black line denoting two extreme scenarios: all patients developing in-hospital MACE and none developing MACE, respectively. Threshold probabilities ranged from 0% to 100% for the training set and 0% to 90% for the testing set (Figs. 5A, 5B). Across this wide range of threshold probabilities, the nomogram consistently provided greater net benefits compared to the extreme scenarios, underscoring its clinical relevance.

## DISCUSSION

The increasing annual prevalence and mortality rates of ACS in young individuals are significantly impacting their quality of life due to high rates of mortality and disability. However, previous studies have primarily focused on assessments of individual factors and their associations with risk. No detailed analyses have been conducted specifically within young populations. This study is the first to develop a nomogram model for predicting the occurrence of in-hospital MACE in young ACS patients.

Our novel nomogram model utilized LASSO regression, effectively managing high-dimensional data and mitigating overfitting, which enhances precision in variable selection compared to conventional models. The proposed nomogram graphically delineates the relative contributions of individual predictors to MACE risk quantification, providing clinicians with a pragmatic bedside tool for personalized risk stratification.

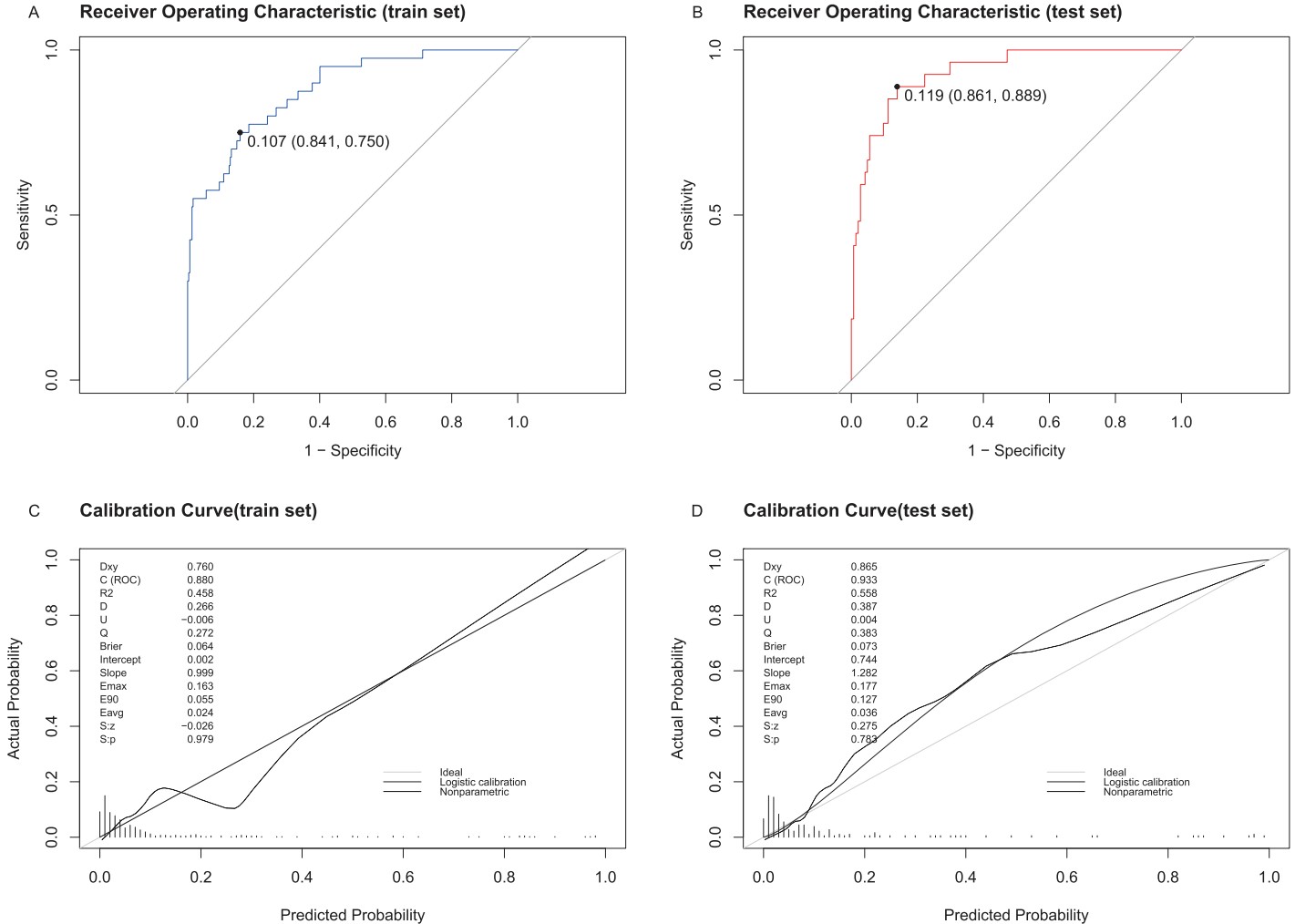

**Figure 4 ROC and the calibration curve of the model were generated to predict in-hospital MACE.** (A) ROC curve and cut-off value for the training set. (B) ROC curve and cut-off value for the testing set. (C) Calibration curve for the training set. (D) Calibration curve for the testing set, showing predicted probabilities on the X-axis and actual probabilities on the Y-axis, with diagonal alignment indicating good calibration.

Beyond static risk categorization, this instrument may facilitate dynamic risk stratification through patient-specific parameter adjustments, allowing real-time recalibration in response to evolving clinical status.

Notably, the testing set demonstrated enhanced discriminative capacity to identify both overt and subclinical high-risk profiles, achieving subclinical high-risk profiles, achieving the AUC was 0.8346 (95% CI [0.742–0.9272]) with a cut-off value of 0.119 (Fig. 4). The dual capability of integrating anatomical severity with metabolic-inflammation markers addresses critical gaps in existing risk paradigms for young ACS patients. We considered admission indicators such as WBC, Killip classification, LYMPH, HR, TG, and the Gensini score, creating a robust model with strong discrimination and calibration. Below, we systematically discuss the clinical relevance of these predictors.

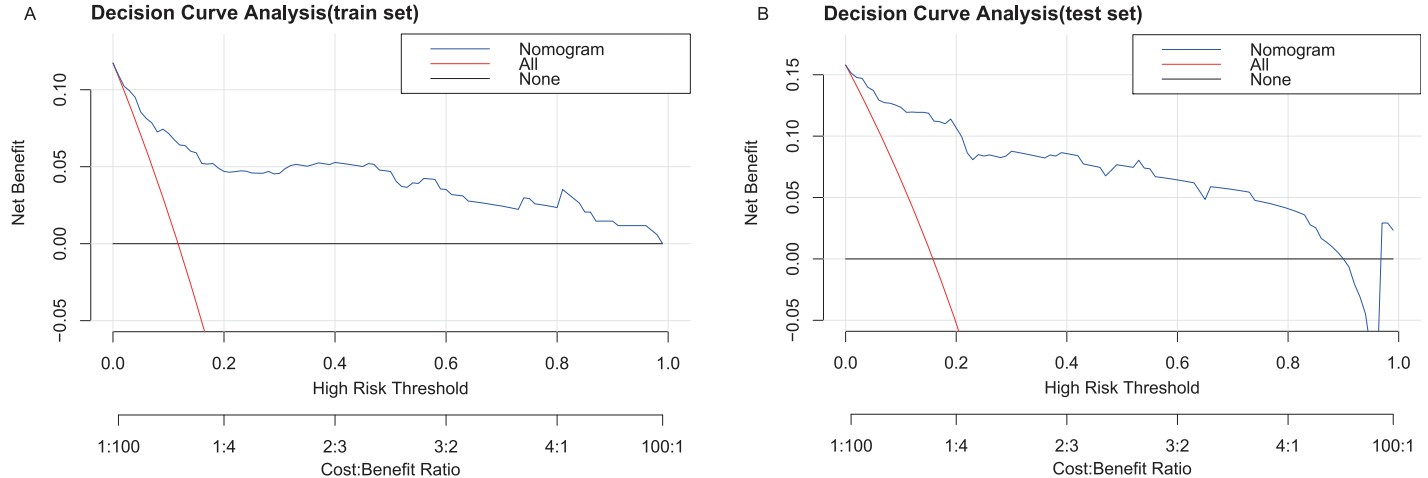

**Figure 5 Decision curve analysis for the clinical utility of the nomogram prediction model.** (A) DCA for the training set. (B) DCA for the testing set. The red line represents the assumption that all patients develop in-hospital MACE, while the black line assumes none do.

## Inflammatory profiling in MACE prediction

WBC count and its subtypes serve as markers of the inflammatory response. They are cost-effective and widely accessible hematological indicators that have been linked to the likelihood of cardiovascular complications such as myocardial infarction and stroke (*Zia et al., 2012*; *Wang et al., 2018*). Our study also revealed that WBC count can be used to predict the occurrence of MACE during hospitalization in young ACS patients. This finding aligns with the results of another study that indicated a higher WBC count was associated with increased in-hospital mortality in patients with ACS (*González-Pacheco et al., 2019*). However, the aforementioned studies did not categorize ACS patients based on age. While the studies mentioned primarily assessed WBC count as a correlational risk factor, our model innovatively assigns individualized prognostic weights to WBC based on its predictive contribution to in-hospital MACE risk. This novel scoring framework, quantified through multivariable LASSO regression, precisely delineates the pathophysiological contribution of leukocytes to acute cardiovascular outcomes—a refinement that is unattainable through conventional univariate analyses. Several researchers have identified a correlation between increased coronary thrombus burden and elevated neutrophil and WBC counts in non-ST elevation ACS patients (*Özkan et al., 2022*). These results help to elucidate the possible pathophysiological mechanisms by which WBC count predicts concomitant MACE in patients with ACS; however, the specific mechanisms involved have not yet been fully identified and warrant further exploration in future.

Our study provides novel insights into immune dysregulation in ACS, demonstrating that LYMPH serves as a pivotal predictor of in-hospital MACE in young patients. This finding complements emerging evidence from elderly cohorts, where elevated LYMPH independently predict the development of heart failure in ACS patients (*Wen et al., 2023*).

While previous studies have shown an elevated NLR in young patients with ACS (*Oztürk et al., 2013*), no prior investigations have systematically evaluated the independent prognostic value of LYMPH itself within this population. In our multivariate logistic regression model, LYMPH demonstrated a clinically meaningful association with MACE risk (OR = 1.18, 95% CI [0.87–1.59]), although statistically nonsignificant ($p = 0.291$). This lack of statistical significance may be attributed to its intricate interplay with other inflammatory markers, such as WBC count (OR = 1.19, $p = 0.002$). This nuanced relationship underscores the dynamics of LYMPH as a complementary biomarker that captures distinct aspects of immune activation beyond conventional inflammation metrics. Notably, our univariate analysis revealed a robust independent association between elevated LYMPH and MACE risk (OR = 1.37, 95% CI [1.09–1.72], $p = 0.007$), aligning with mechanistic evidence of lymphocyte-mediated plaque destabilization. CD8+ T-cell subsets (*e.g.*, CD57+ cells) are known to drive cytokine storms in ruptured plaques (*Brunetti, 2014*), while CD4/CD28-null T-cell infiltration exacerbates plaque vulnerability (*Dumitriu, 2015*). Our findings extend these observations by demonstrating that LYMPH serves as a surrogate for adaptive immune hyperactivity in young ACS patients, a population prone to exaggerated inflammatory responses.

### Anatomical severity in MACE prediction

The Gensini score is computed for individual lesions by considering factors such as severity score, lesion area, and coronary collateral modulation factor, with the total score being the sum for each coronary lesion (*Rampidis et al., 2019*). This scoring system has a critical advantage in capturing the complex lesion patterns characteristic of youth-onset ACS, whereas the Synergy between Percutaneous Coronary Intervention with Taxus and Cardiac Surgery (SYNTAX) score primarily focuses on revascularization strategy planning, and Society for Cardiovascular Angiography and Interventions Shock Classification (SCAI) lacks anatomical granularity. Recent findings indicate a significant association between the Gensini score and all-cause mortality in elderly patients with ACS and diabetes mellitus (*Shen et al., 2023*). Additionally, reports have shown that a high Gensini score is an independent risk factor for MACE in patients with ACS (*Shi et al., 2019*). However, few studies have examined whether the Gensini score is a suitable predictor for the development of MACE in young ACS patients. To the best of our knowledge, our prediction model demonstrates for the first time that the Gensini score is an important factor in predicting MACE in young patients.

### Hemodynamic-metabolic indicators in MACE prediction

HR is one of the important predictive indicators for cardiovascular disease, particularly in patients with ACS. In a predictive model for assessing the risk of 30-day mortality in AMI patients, researchers developed an individualized scoring system that evaluates patient prognosis by quantifying the contributions of various risk factors, with HR emerging as a critical prognostic determinant (*Suryawan et al., 2025*). This finding is consistent with the results of our study. However, significant demographic differences exist between the study cohorts. The referenced research population demonstrated a mean age of 69.86 ± 9.47

years, whereas our cohort comprised notably younger patients with a median age of 38.0 years. A study indicated that an elevated HR on admission was linked with significantly increased risk of MACE in ACS patients. Interestingly, this relationship was observed in AMI patients without low LVEF, but not in patients with low LVEF (<40%) (*Zhang et al., 2023*). This is consistent with our research findings; However, our research population was expanded to UA. Studies have shown that whether in patients with ACS or chronic coronary syndrome, the risk of MACE significantly increases when the patient's HR was greater than or equal to 75 beats per minute (*Oba et al., 2023*). In our study, we also found that the HR of ACS patients in the MACE group was 85.0 (70.5–99.5) beats per minute. However, the mechanism by which HR affects prognosis of ACS is not clear.

Currently, it was widely believed that elevated levels of LDL-C play an important role in the pathogenesis of CAD (*Chen et al., 2024*). However, in recent years, there has been a growing recognition that elevated cholesterol levels of TG-rich lipoprotein particles contribute to ischemic heart disease (*Varbo et al., 2013*). Moreover, reducing TG could achieve some clinical benefits (*Ference et al., 2019*). The study found that TG as a major component of small-density LDL-C combined with body mass index could stratify risk in patients with ACS (*Hori et al., 2021*). Elevated TG was also one of the factors promoting the occurrence of ACS (*Islam et al., 2012*). In our study, TG was also found to be one of the important factors in predicting the occurrence of MACE in young ACS patients. However, reports have found that patients with ACS accompanied by low TG have an increased risk of all-cause mortality at 6 months and 3 years (*Khawaja et al., 2011*). Therefore, the role of TG in ACS still needs further research.

## Killip classification in MACE prediction

The Killip classification is a quick and straightforward clinical tool used for risk stratification and is an effective method for assessing the risk of patients with AMI and for determining patient prognosis (*Hashmi et al., 2020*). A previous study indicated that a Killip classification >1 independently predicts in-hospital mortality, stroke, and myocardial reinfarction in cases of nonpremature age presenting with ACS (*Menezes Fernandes et al., 2022*). Our findings are consistent with the results of these aforementioned studies, especially those that identify a Killip class ≥2 as an independent predictor of in-hospital MACE (OR: 18.69, 95% CI [6.63–52.71], $p < 0.001$) in young patients with ACS. It was discovered that Killip classifications III and IV were both independent predictors of in-hospital mortality and MACE during follow-up (*Del Buono et al., 2021*; *Bernardus, Pramudyo & Akbar, 2023*). However, the limited number of patients with Killip class III and IV in our cohort constrained the statistical power for stratified analyses across Killip subcategories and then we lack long-term follow-up data.

Our analysis revealed a striking male predominance in disease prevalence (95.1%), which is consistent with previous epidemiological reports (*Alfaddagh et al., 2020*). This gender disparity may be multifactorial, influenced by higher smoking rates among males (*Papathanasiou et al., 2024*) and delayed disease onset in females (*Dey et al., 2009*). Cigarette smoking constitutes a critical modifiable risk factor for both premature onset of ACS (*Mahendiran et al., 2023*) and subsequent MACE in ACS patients (*Okkonen et al.,*

2021*). However, in our study, smoking was not included as a predictive variable in the model. This omission may be attributed to the fact that smoking can mediate the risk of MACE through downstream mechanisms, such as systemic inflammatory responses (*Pedersen et al., 2019*) and metabolic dysfunction (*Li et al., 2022*). Consequently, its independent predictive value for MACE may be diminished by the stronger associations of other variables within these downstream mechanisms.

## Comparative model performance

The Global Registry of Acute Coronary Events Risk Score (GRACE) score (*Elbarouni et al., 2009*), while widely validated, disproportionately emphasizes age, resulting in underestimated risk stratification for younger populations due to minimal age-related weighting. Furthermore, GRACE overlooks metabolic (*e.g.*, TG) and anatomical factors (*e.g.*, Gensini score), which are pivotal in younger cohorts with evolving plaque vulnerability and metabolic dysregulation. Similarly, the Canada Acute Coronary Syndrome (C-ACS) (*Huynh et al., 2013*) and Portuguese Registry of Acute Coronary Syndromes (ProACS) (*Timóteo, Mimoso & em nome dos investigadores do Registo Nacional de Síndromes Coronárias Agudas, 2018*) models prioritize hemodynamic parameters (*e.g.*, BP, HR) and clinical classifications (*e.g.*, Killip classification) but lack integration of objective anatomical severity or inflammatory biomarkers. This omission is particularly consequential in young patients, where subclinical inflammation and coronary complexity often drive acute outcomes.

The PADjadjaran Mortality in Acute coronary syndrome (PADMA) (*Pramudyo et al., 2022*) and HEART scores (*Frisoli et al., 2017*), though clinically practical, rely heavily on subjective assessments (*e.g.*, patient history) or narrowly focus on mortality rather than composite MACE. In contrast, our model leverages data-driven LASSO regression to identify young-specific predictors, including TG (metabolic risk), Gensini score (quantifying coronary burden), and WBC/LYMPH (inflammation response). These variables collectively address the multifactorial pathophysiology of ACS in younger individuals, where traditional age-centric models falter. Compared to established risk models, the current nomogram demonstrates distinct advantages in predicting in-hospital MACE among young ACS patients by addressing critical limitations of existing tools.

Notably, the inclusion of Gensini score and LYMPH represents a novel advancement. While prior studies linked these factors to long-term outcomes in older populations, our model establishes their prognostic relevance for in-hospital MACE in younger patients—a population historically underrepresented in ACS risk stratification.

Study Limitations:

(i) The retrospective design may introduce selection bias and unmeasured confounders, necessitating prospective validation.

(ii) Single-center studies with small samples require validation in larger, diverse multicenter populations to confirm generalizability.

(iii) Single-timepoint laboratory indicators measurements limit dynamic risk assessment; serial monitoring could enhance prognostic precision.

(iv) Long-term follow-up is needed to assess post-discharge outcomes and chronic prognosis.

(v) We did not account for patients' medication history in this study, thereby potentially overlooking confounding effects of pharmacological interventions on the observed outcomes.

To summarize, we showed that the model may be helpful in identifying high-risk patients being evaluated for young ACS patients who are most likely to develop MACE in hospital. With this model, clinicians need to focus on the WBC, Killip, LYMPH, HR, TG and Gensini score.

## CONCLUSIONS

WBC, Killip, LYMPH, HR, TG and Gensini score are independent predictors of in-hospital MACE for young patients with ACS. The nomogram model, developed using the factors mentioned above, demonstrated strong discriminatory and calibration abilities. It assists in predicting the risk of in-hospital MACE for young ACS patients.

### Funding
The authors received no funding for this work.

### Competing Interests
The authors declare that they have no competing interests.

### Author Contributions
- Jia Zheng conceived and designed the experiments, performed the experiments, analyzed the data, prepared figures and/or tables, authored or reviewed drafts of the article, and approved the final draft.
- Junyang Li conceived and designed the experiments, performed the experiments, analyzed the data, prepared figures and/or tables, authored or reviewed drafts of the article, and approved the final draft.
- Tingting Li performed the experiments, analyzed the data, prepared figures and/or tables, and approved the final draft.
- Fang Hu performed the experiments, prepared figures and/or tables, resources, and approved the final draft.
- Degang Cheng performed the experiments, authored or reviewed drafts of the article, and approved the final draft.
- Chengzhi Lu conceived and designed the experiments, performed the experiments, analyzed the data, authored or reviewed drafts of the article, and approved the final draft.

### Human Ethics
The following information was supplied relating to ethical approvals (*i.e.*, approving body and any reference numbers):

Research was approved by Tianjin First Central Hospital (approval No 20240515-1).

## Data Availability

The data are available in the Supplemental File.

## Supplemental Information

Supplemental information for this article can be found online at http://dx.doi.org/10.7717/peerj.19513#supplemental-information.

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
