# Peer review of "Development and validation of an in-hospital major adverse cardiovascular events risk model for young patients with acute coronary syndrome: a retrospective cohort study"

_PeerJ, doi:10.7717/peerj.19513_

## Round 0.1 · original submission · Major Revisions

Four reviewers have commented. Please address their concerns in an appropriate revision

·

Basic reporting

no comment

Experimental design

no comment

Validity of the findings

The study focuses only on in-hospital MACE. I think long-term outcomes should be assessed, which could provide additional insights into the predictive value of the model.

Additional comments

The study population consists of 95.1% males. I think the figure should be confirmed again. Is the incidence of ACS so different between males and females?

·

Basic reporting

- There are several grammatical errors need to be addressed by the authors. I suggest the author to use professional english language service to tackle this issue
- Literature references were not sufficient, please add more references and make it Vancouver style
- If you do not mind, please add new literature (https://doi.org/10.20944/preprints202412.2252.v1) about scoring system for AMI mortality in Asian population

Experimental design

- Please explain how did you limit age 45 years for young age? Which references you have used?
- For LASSO regression, please kindly explain which LASSO you have used? Since it is confusing for the readers and reviewers to address the validity of your test
- Please explain the role of Gensini score for this predictor. Whether it could be replaced with Syntax score or SCAI class?

Validity of the findings

- Since inhospital MACE might be biased between observers, I wonder what kind of test and measurement to ensure the validity and reliability for your study.
- Also number of your subjects is very small thus limiting the generalisability of your study

Additional comments

Fine

Reviewer 3 ·

Basic reporting

The incidence of acute coronary syndrome (ACS) in younger populations is increasing annually. In this study, the authors develop a nomogram model to predict the risk of in-hospital major adverse cardiovascular events (MACE) in young ACS patients based on multiple diagnostic indicators. The study demonstrates good novelty and clinical significance.

Experimental design

1. The authors should provide a detailed description of the methodology used in this study, including the criteria for selecting participants, the assessment indicators and methods, the observed outcomes, and how the training and testing sets were chosen. A simple flowchart is insufficient to convey the key information about the research process.
2.The authors should clarify the rationale behind selecting these clinical indicators. What was the basis for choosing these particular indicators for analysis?

Validity of the findings

The descriptions for Table 2 and Figures 2 and 3 are overly brief. It would be helpful to provide more detailed and comprehensive descriptions of the results.

Additional comments

The authors should reorganize the discussion to streamline and consolidate the content related to different indicators. Additionally, the authors should discuss the differences and advantages of the predictive model proposed in this study compared to previous similar models. It is also important to include a discussion of the limitations of the current study.

Reviewer 4 ·

Basic reporting

Clear and unambiguous, professional English used throughout.
Literature references, sufficient field background/context provided.
Professional article structure, figures, tables. Raw data shared.
Self-contained with relevant results to hypotheses.

Experimental design

Original primary research within Aims and Scope of the journal.
Research question well defined, relevant & meaningful. It is stated how research fills an identified knowledge gap.
Rigorous investigation performed to a high technical & ethical standard.
Methods described with sufficient detail & information to replicate

Validity of the findings

Impact and novelty not assessed. Meaningful replication encouraged where rationale & benefit to literature is clearly stated.
All underlying data have been provided; they are robust, statistically sound, & controlled.
Conclusions are well stated, linked to original research question & limited to supporting results.

Additional comments

1. How does your study differ from previous literature examining mortality risk factors in young ACS patients? A more detailed discussion of the novel aspects of your nomogram model would strengthen the manuscript.
2. How were the training and test groups assigned? Were they randomly assigned or were specific criteria applied? Explaining the methodology used for data partitioning would increase the transparency of the study.
3. How does your model compare to previously developed ACS risk prediction models? Including a comparative analysis of model performance measures would increase the clinical relevance of your findings.
4. How was the effect of smoking controlled for in the multivariable model? It would be useful to clarify whether smoking was treated as an independent predictor and how it affected the final model.
5. Was the patient's medication history taken into account in the analysis? Given the potential impact of antiplatelet agents, statins, and beta-blockers on MACE outcomes, it would be valuable to discuss their role in model accuracy and risk stratification. Or should they be included in the limitations.

---

## Round 0.2 · accepted · Accept

Thank you for comprehensively addressing all of the reviewer comments, which have significantly improved this manuscript.

·

Basic reporting

no comment

Experimental design

no comment

Validity of the findings

no comment

Additional comments

It is indeed a pleasure to review the revised manuscript. The authors have made significant improvements to the manuscript. In this study, They made a great effort to develop and validate a risk prediction model for in-hospital major adverse cardiovascular events (MACE) in young acute coronary syndrome (ACS) patients. This study focuses on the incidence of MACE in young ACS patients. Using a retrospective analysis of 513 young patients, the study employs statistical methods, including LASSO regression and logistic regression, to identify key risk factors. The developed nomogram model, validated through ROC curves and decision curve analysis, demonstrates strong predictive accuracy.

·

Basic reporting

Clear, unambiguous, professional English language used throughout

Experimental design

Research question well defined, relevant & meaningful

Validity of the findings

All underlying data have been provided; they are robust, statistically sound, & controlled

Additional comments

Fine and well defined

Reviewer 3 ·

Basic reporting

-Clear and unambiguous, professional English used throughout.
-Literature references, sufficient field background/context provided.

Experimental design

-Original primary research within Aims and Scope of Peer J.
-Research question well defined, relevant & meaningful.

Validity of the findings

-Conclusions are well stated, linked to original research question & limited to supporting results.

Additional comments

The authors have made significant improvements. Therefore, I have no further comments.